# The Indiscriminate Chemical Makeup of Secondary Metabolites Derived from Endophytes Harvested from *Aloe barbadensis* Miller in South Africa’s Limpopo Region

**DOI:** 10.3390/molecules29061297

**Published:** 2024-03-14

**Authors:** Mpho Mamphoka Nchabeleng, Thierry Youmbi Fonkui, Green Ezekiel

**Affiliations:** Department of Biotechnology and Food Technology, Faculty of Science, University of Johannesburg, Doornfontein Campus, Johannesburg 2028, South Africa; mphon@uj.ac.za (M.M.N.); thierryfy@uj.ac.za (T.Y.F.)

**Keywords:** endophytes, secondary metabolites, *Mycobacterium bovis*, tuberculosis, medicinal plants

## Abstract

The efficacy of 23 bacterial isolates obtained from surface-sterilized stems and leaves of three medicinal plants (*Aloe barbadensis* Miller, *Artemisia afra*, and *Moringa oleifera*) was investigated in an endeavour to prevent the growth of *Mycobacterium bovis* using the cross-streak method. Endophytes were isolated by incubating sterile plant materials on nutrient agar at 30 °C for 5 days. Two isolates showing activity were subsequently utilized to produce the extracts. Whole-genome sequencing (WGC) was used to identify the isolates. Secondary metabolites produced after 7 days of growth in nutrient broth were harvested through extraction with ethyl acetate. The extracts were chemically profiled using gas chromatography–high resolution time-of-flight mass spectrometry (GC–HRTOF-MS). NCBI BLAST search results revealed that the isolated endophytes belonged to the Pseudomonas and Enterobacter genera, based on WGC. Two endophytes, Aloe I4 and Aloe I3–I5 from *Aloe barbadensis*, exhibited potency based on the cross-streak method. The metabolite profiling of the selected endophytes identified 34 metabolites from Aloe I4, including ergotamine, octadecane, L-proline and 143 other metabolites including quinoline and valeramide, which inhibit microbial quorum sensing. These findings suggest that bacterial endophytes from medicinal plants, particularly *Aloe barbadensis,* hold promise as sources of antimycobacterial agents for human health applications.

## 1. Introduction

The healthcare sector has become greatly concerned with the global growing resistance to antimicrobial and chemotherapeutic medications in recent years. There is a pressing need to explore and create novel anticancer and antimicrobial agents [1]. As a result, there has been a renewed passion for investigating microorganisms as a potential repository of new discoveries. This is based on the remarkable success of bacterial compounds in the production of effective antimicrobial agents, anticancer drugs, and agricultural pesticides [2]. As an alternative to antibiotics, bacterial bioactive chemicals may play a role in microbe-to-microbe and microbe-to-host reactions, according to a growing body of research [3].

The use of medicinal plants to address human health problems dates back many years, and the benefits of medicinal plants in the health care system are undeniable. Plants are important and reliable reservoirs of bioactive compounds, as evidenced by their prolonged and continuous use in addressing the pressing issues in human health [4]. *Aloe barbadensis* Miller, commonly known as aloe vera, *Artemisia afra*, known as wormwood, and *Moringa oleifera*, generally named drumstick tree, are three important medicinal plants used in South Africa to address various clinical conditions. Found in tropical and temperate regions, aloe vera is traditionally used for the treatment and prevention of various ailments such as sunburns, wounds, skin disorders, diabetes, ulcers, and other conditions [5]. Recent pharmacological data indicated that Aloe vera displayed anticancer action, digestive protective activity, and antimicrobial properties [6]. *Artemisia afra*, on the other hand, is commonly used as an infusion for the treatment of malaria and ailments such as colds, fever, influenza, sore throats, pneumonia, and many other ailments [7,8]. The antibacterial activity of wormwood against *Mycobacterium tuberculosis* has been documented [9]. *Moringa Oleifera* is another important medicinal plant harbouring bioactive substances in its seeds, leaves, flowers, and pods that is used to alleviate symptoms such as joint pains, malnutrition, and headaches, with many applications in the food sector [10]. *Moringa* is considered an effective agent against hypocholesterolaemia and hypolipidemia and has also shown antimicrobial properties against various human pathogens including *E. coli* and *S. aureus* [11]. Medicinal plants, like any other plants, are generally susceptible to abiotic and biotic stress conditions. To manage and control these attacks, plants often draw benefit from their symbiotic relationship with microorganisms found in their tissues [12]. Endophytes (bacteria and fungi) are suitable suppliers of bioactive chemicals that do not only benefit these plants but have also found use in disease control.

Bacterial endophytes reside within plants’ internal tissues without any notable signs of negative effects on the host. Endophytes have characteristics which are beneficial to their host plants, such as promoting development, maintaining protection against insects and pests, exhibiting antimicrobial properties against plant pathogens, and aiding in regulating stress. They facilitate global nutrient cycling and the differentiation between host health and disease states [13]. They also serve as valuable sources of biotechnologically significant molecules, with numerous biocompatible and therapeutic substances. These organisms function as reservoirs for distinct bioactive metabolites, including alkaloids, phenolic acids, steroids, and tannins, with antibacterial, insecticidal, and anticancer properties, [14] to name a few.

Secondary metabolites play a crucial role in mediating microbial interactions in natural environments, exhibiting a wide range of activities which can span from competitive to cooperative dynamics. Instances of such interactions encompass the synthesis of bioactive secondary metabolites. It is worth noting that these metabolites exhibit variable degrees of specificity, wherein certain metabolites exhibit a broad-spectrum effect by targeting a large range of molecular targets, while others exhibit a narrow-spectrum effect by targeting a more limited scope of molecular targets [4]. The exploration of endophytic secondary metabolites with bioactive properties has gained increasing attention, owing to their various bioactive capabilities and wide range of structural categories. Currently, only a few plants have been studied to explore their endophytic biodiversity and ability to produce bioactive secondary metabolites. In effect, every medicinal plant species on the planet is a habitat for multiple varieties of endophytic bacteria. Endophytes can undertake species-specific interactions and co-evolve with their plant hosts.

The best-known medicinal usage of *Aloe barbadensis* Miller, a succulent plant which thrives in dry and subtropical climates, is in Ayurvedic, homoeopathic and allopathic medicine. Throughout history, diverse nations have used aloe vera extensively for various purposes. These include using aloe vera to decrease perspiration, administering the plant orally to manage diabetes, and employing the plant to alleviate a variety of gastrointestinal disorders. It has been utilized for managing burn injuries, minor lacerations, genital herpes, and eczema. The leaves of this medicinal plant are abundant in vitamins, minerals, natural sugars, enzymes, amino acids, and various bioactive molecules. These molecules possess soothing, aperient, anti-inflammatory, antioxidant, antimicrobial, antihelmenthic, antifungal, aphrodisiac, antiseptic and cosmetic properties. Given its healing and rejuvenating properties, cosmetic companies use this plant widely. This study aimed to screen, identify, and characterize the secondary metabolites associated with two bacterial endophytes originating from native medicinal plants in Limpopo, South Africa.

## 2. Results

In this study, twenty-three (23) bacterial endophytes were isolated from the stems and leaves of the abovementioned medicinal plants. Seven (7) bacterial endophytes were isolated from *Moringa oleifera*, nine (9) from *Artemisia afra* and seven (7) from *Aloe vera*. The isolates were then screened for antimicrobial properties, and the chemical profiles of biologically active isolates were studied and are presented below.

### 2.1. Antimicrobial Susceptibility of Mycobacterium bovis

The cross-streak method used to measure antimicrobial activity revealed that *M. bovis* was sensitive to secondary metabolites produced by some bacterial isolates. Of the 23 endophytic bacteria tested, 21 showed no activity against the test organism *M. bovis*, while two isolates, Aloe I4 and Aloe I3–I5, showed clear areas of inhibition against *M. bovis* (Figure 1).

An inhibitory zone of 5 mm was recorded with isolate Aloe I4, and no growth of *M. bovis* was observed in the Petri dish treated with Aloe I3–I5 (Figure 2). The complete absence of growth or total inhibition noted here could be attributed to the chemical profile of the isolate.

Our findings corroborate similar observations in the literature [14]. Previous studies have proven that endophytes produce active compounds which exhibit antibacterial, antifungal, antiviral, antisuppressant, and antioxidant properties [15]. According to Lertcanawanichakul and Sawangnop [16], the cross-streak method is preferred over the agar well diffusion method, but it can have drawbacks as the margins of the inhibition zones are usually indistinct, making it difficult to obtain quantitative data. This was experienced in this study; however, with the clear zones/areas observed on the plates containing Aloe I4 and Aloe I3–I5, compared to the other 21 isolates (Figure 2), it could be deduced that they have antimicrobial activity against *Mycobacterium bovis*. Other methods, such as the gel well diffusion method, could be employed to further characterize the isolates’ bioactivity against test species. Endophytes and medicinal plants produce vital secondary metabolites which could be used for human benefit.

### 2.2. Isolation and Characterization of Endophytes

Surface sterilization aids in the removal of potential pollutants (such as soil) and the promotion of internal microorganismal growth [17]. As no bacterial colonies were seen on any of the control plates, the surface sterilization of all plant parts was deemed successful. Various colonies were observed on the plates, with pigmentation varying between white, cream white, yellow, pinkish, and pale yellow. The endophytic isolates varied in texture between viscid, pasty, moist, mucoid, and dry. In this study, twenty-three (23) bacterial endophytes were isolated from the stems and leaves of the abovementioned medicinal plants. Seven (7) bacterial endophytes were isolated from Moringa oleifera, nine (9) from Artemisia afra, and seven (7) from Aloe vera. Our findings corroborate those of Strobel [18], who reported the successful isolation of endophytes from plant parts such as roots, leaves, and stems of Rhyncholacis penicillata. Microscopy analysis revealed thirteen (13) Gram-negative rods and ten (10) Gram-positive rods and cocci isolates. Host plant species, maturity, territorial and ecological distribution, season of sample collection, exterior sterilization method, and growth conditions are all variables which can impact the diversity and dispersion of bacterial endophytes in plants [19]. Several factors influence the distribution and variety of bacterial endophytes in plants, including host plant species, age, plant tissue type, geographical and habitat distribution, sampling season, surface sterilization method, growth media, and culture conditions [19]. The morphological characteristics mentioned here are insufficient for successfully identifying and characterizing bacterial endophytes or species, so molecular techniques such as whole-genome sequencing (WGS) were used to fully identify the species in terms of phylogeny, morphology, virulence, and other important factors [20].

### 2.3. Molecular and Phylogenetic Identification of Aloe I4 and Aloe I3–I5

Whole-genome sequence (WGS) analysis has become an important tool for determining bacterial relationships and is commonly utilized for the identification of microorganisms. It is used frequently to determine phylogenetic relationships between organisms [21]. De novo genome assemblies were used in this project for the molecular and phylogenetic identification of the isolates. According to the NCBI BLAST results for the Aloe I4 and Aloe I3–I5 gene sequences, the isolates belong to two different bacterial genera: *Pseudomonas* and *Enterobacter*. The WGS results of the isolates (Table 1 and Table 2) were submitted to the NCBI database for genome sequences. *Enterobacter* sp. I4, the whole genome shotgun project, was deposited on GenBank (https://DDBJ/ENA/GenBank (accessed on 27 March 2022)) and assigned a Bio-sample number SAMN26660210 and Bio-project number PRJNA816151 under the accession JALBUO000000000. *Pseudomonas* sp. I3–I5, the whole genome shotgun project, was deposited on GenBank (https://DDBJ/ENA/GenBank (accessed on 27 March 2022)) and assigned a Bio-sample number SAMN26660159 and Bio-project number PRJNA816147 under the accession CP093940.

According to Menpara and Sumitra [22], dominating endophytes, particularly in medicinal plants, belong to genera such as *Enterobacter*, *Acinetobacter*, *Staphylococcus*, and *Pseudomonas*. The two bacterial endophytes identified in this project belonged to the *Enterobacter* and *Pseudomonas* genera. The BLAST results showed that the bacterial endophyte Aloe I4 has 100% similarity to several *Enterobacter* species, but the top hit was *Enterobacter cloacae* ATCC 13047. Other species included *Enterobacter mori* LMG 25706, *Enterobacter bugandensis* EB-247, and *Enterobacter dykesii* EIT. It also showed 92% similarity to *Enterobacter hormaechei* subsp. And *Enterobacter Xiangfangensis* LMG 27195. Two outgroup species that are closely related to *Enterobacter* were also revealed, namely *Klebsiella quasipneumoniae* 01A030 and *Kluyvera cryocrescens* NBRC 102467 (Figure 3). *Enterobacter cloacae (E. cloacae)* can be found in numerous places, such as sewage, soil, and food [23]. *E. cloacae*, *E. aerogenes* and *E. sakazakii* are species that are commonly present in clinical materials. *Enterobacter* colonies are either non-pigmented or yellow pigmented [24]. In this project, the colonies of Aloe I4 appeared yellow pigmented. Due to the 100% phylogenetic similarity of Aloe I4 and *Enterobacter cloacae* ATCC 13047, it was assigned the strain name *Enterobacter cloacae* I4.

*Pseudomonas* species are frequently found in plants and have been isolated from a variety of plant parts and tissues [25]. Isolate Aloe I3–I5 showed 99% similarity to *Pseudomonas fulva* DSM 17717 and *Pseudomonas parafulva* DSM 17004. It also showed 85% similarity to *Pseudomonas sichuanensis* WCHPS060039, 70% similarity to *Pseudomonas reidholzensis* CCOS 865, and 63% similarity to *Pseudomonas putida* NBRC 14164 (Figure 4). Because Aloe I3–I5 showed a very close relation to *Pseudomonas fulva* DSM 17717, it was assigned a strain name of *Pseudomonas fulva* I3–I5.

### 2.4. Chemical Profile of Aloe vera I3–I5 and Aloe vera I4

Plants and endophytes produce bioactive chemical substances which could have industrial applications, including pharmaceuticals, cosmetics, agriculture, and food and beverages [26]. In this project, GCMS was used to identify secondary metabolites produced by endophytic isolates with antimicrobial activity against *Mycobacterium bovis*. Both isolates, namely, *Pseudomonas fulva* I3–I5 and *Enterobacter cloacae* I4, produced a variety of active compounds, with functional groups ranging across alkanes, esters, alcohols, ketones, and other organic compounds.

Esters include benzoic acid, n-hexadecanoic acid, propanoic acid, benzeneacetic acid, benzene-propanoic acid, 1.2-Benzenedicarboxylic acid, 4-Hydroxybenzoic acid, and L-proline. Alkanes include tridecane, hexadecane, octadecane, eicosane, and pentadecane. Alcohols include tryptophol, methylalcohol, 2.4-Ditert-butylphenol, 1-octen-4-ol, and phenylethyl alcohol. Ketones include ergotamine, 1-methyl-2-pyrollidinone, and 2-pentanone. Other organic compounds include indole, acetaldehyde, pyridine, 1-hexadecene, bisacrylamide, 2.6-octadiene, bis (2-ethylhexyl phthalate, benzohydrazide, and crotetamide) (Table 3 and Table 4). Endophytes produce a diverse spectrum of metabolites with a variety of bioactivities. These chemical metabolites are beneficial in drug development due to their bioactive effects such as antibacterial, antifungal, immunosuppressive, anticancer, and antioxidant properties, as previously mentioned. It is due to these important functions that endophytes are gaining popularity in terms of research, as they can be employed to treat various diseases which are currently difficult to medicate, owing to the ubiquitous global drug resistance. Examples of active compounds isolated from both *Pseudomonas fulva* I3–I5 and *Enterobacter cloacae* I4 ethyl acetate extracts with pharmaceutical or medicinal applications include hexadecane, pyrollo, ergotamine, L-proline, octacosane, phenylethyl alcohol, 1-Methyl-2-Pyrrolidinone, and benzenepropanoic acid.

Hexadecane was one of the active compounds produced by both isolates and has been proven to have antifungal, antibacterial, and antioxidant properties [27]. Symptoms of human tuberculosis caused by *M. bovis* infection include fever, headaches, heavy sweat, coughing, abdominal pain, weight loss, and diarrhoea [28]. According to [29], ergotamine has vital properties such as antibacterial and antifungal activity. It is also employed in pharmaceuticals for the treatment of fever, headaches, and migraine. Fever is one of the symptoms of TB, and this active compound in isolation could be employed and incorporated in the treatment of TB. Pyrollo was isolated from ethyl acetate extracts of *Pseudomonas fulva* I3–I5 and *Enterobacter* cloacae I4, which is used in anti-inflammatory drugs, antibiotics and antitumour agents [30]. Methyl alcohol was extracted from *Enterobacter cloacae* I4 and is known to possess antimicrobial properties. It is commonly employed in pharmaceuticals to produce cholesterol, as well as antibiotics such as streptomycin and vitamins and hormones.

Another active compound extracted from *Pseudomonas fulva* I3–I5 and *Enterobacter cloacae* I4 with bioactive properties was octacosane, which is used in the synthesis of proteins [31]. L-proline was identified in both isolates; it is known to have antibacterial and antifungal properties and is used in pharmaceutical preparations such as injections [32]. Most of the active compounds extracted in this study, such as bifenthrin, bumetrizole, nonanal, metolachlor, and quinolone, have pharmaceutical applications but could also have other biotechnological significances, such as uses in cosmetics, pesticides, and detergents, as well as in food and beverages. This study proves the significance of employing medicinal plants and endophytes in drug discovery and development, to eradicate or decrease the rising microbial resistance taking place globally. Endophyte isolation and the investigation of their active compounds is gaining traction worldwide due to their vital health properties [33].

## 3. Materials and Methods

Three medicinal plants, namely, *Aloe barbadensis* Miller (Aloe vera), *Artemisia afra* (Wormwood), and *Moringa oleifera* (Drumstick tree) were harvested from Apel village in Ga-Sekhukhune (Limpopo province, South Africa, 24°24′0″ S, 29°44′0″ E). The plants were collected and placed in sterile polyethylene bags and transported to the University of Johannesburg laboratory at 4 °C.

### 3.1. Antimicrobial Susceptibility of Mycobacterium bovis against Bacterial Endophytes Isolated from Medicinal Plants

#### 3.1.1. Culturing of *Mycobacterium bovis*

*Mycobacterium bovis* (Kwikstik, Microbiologics, 01203P) was cultured in middlebrook 7H9 broth with oleic acid albumin dextrose catalase (OADC) supplement at a temperature of 37 °C for five days, while shaking at 150 rpm.

#### 3.1.2. Susceptibility Test of *Mycobacterium bovis* against Endophytic Bacteria

The cross-streak technique was used to assess the antimycobacterial potency of the 23 isolates against *Mycobacterium bovis* following the method of Okudo and Wallis [34]. In brief, Muller–Hinton agar (MHA) (Sigma-Aldrich, Johannesburg, South Africa, BCBZ7677) plates were prepared following manufacturer guidelines, and 20 mL of the medium was allowed to solidify in Petri dishes. A line was drawn in the centre to divide the plates into twos at the back of each plate for the inoculation of the 23 isolated endophytes. Perpendicularly to this line, two other lines were drawn for the inoculation of *M. bovis*. Using an inoculation loop, a loop full of each of the 23 bacterial isolates were streaked throughout the central line on the Petri dishes containing the solidifies Muller–Hinton agar and incubated at 37 °C for 3 days for maximum growth. Then, afterwards, the plates were cross streaked with a loop full of test organism (*M. bovis*) at a 90° angle (on the two lines drawn prior) to the bacterial isolates and further incubated for 5 days at 37 °C to allow for antimicrobial activity.

Endophytic bacteria were isolated from plant components (fresh, healthy, and disease-free stems and leaves) immediately after therapeutic plants were collected. First, all medicinal plant parts (stems and leaves) were washed under running tap water to eliminate contaminants such as soil and dust debris. Plant components were sliced into small segments and treated with 5% Tween 20 for 5 min while vigorously shaking them, then rinsed with distilled water (dH_2_O) several times [25]. The next step was disinfecting the plant components with 70% ethanol for 1 min, after which components were cleansed five times with dH_2_O to remove any ethanol residue. They were then treated with 1% sodium hypochlorite and rinsed five times with dH_2_O, and the final wash was plated on nutrient agar (NA) as a control. The plant components’ outer surfaces were removed, and they were ground in phosphate-buffered saline (PBS). Before being spread out on nutrient agar plates, all macerated components were serially diluted to a concentration of 10^−3^ [25]. The plates were incubated at 30 °C for 5 days with the controls, with bacterial growth measured daily. After the incubation period, various colonies were selected and sub-cultured on nutrient agar to yield pure isolates. Pure bacterial isolates were kept at −80 °C with 50 percent glycerol on a 1 mL glycerol–1 mL broth medium ratio overnight [25].

The whole-genome sequencing (WGS) was conducted using PacBio technology by Inqaba Biotec in Pretoria, South Africa. The newly generated genome sequences were compared to the most closely related bacterial species using the Basic Local Alignment Search Tool (BLAST) on the NCBI platform (https://blast.ncbi.nlm.nih.gov, accessed on 30 March 2022).

#### 3.1.3. Processing of Samples for Metabolite Profiling

Following the modified approach proposed by Xu et al. [35] and Daji et al. [36], centrifuge tubes were employed to combine 10 mL of 100% methanol with 1 g of the extracts. Using Scientech 704 from Labotech in Johannesburg, South Africa, samples were vortexed before an ultrasonic-aided extraction was carried out for one hour at 4 °C. Subsequently, samples underwent centrifugation at 3500× *g* revolutions per minute (rpm) at a temperature of 4 °C for 5 min using an Eppendorf 5702R centrifuge (Merck, Johannesburg, South Africa). The liquid portion was filtered using filter sheets with a pore size of 0.45 micrometres and transferred into dark amber vials. The extraction process was performed in triplicate for each sample.

#### 3.1.4. Analysis Using GC-HRTOF-MS

To achieve a reproducible result, the mass optimization of the instrument (Leco Pegasus GC–HRTOF-MS, St. Joseph, MI, USA) was performed and passed the preceding analysis. The extracts were introduced into the GC–HRTOF-MS system (Gerstel GmbH & Co. KG, Mülheim (Ruhr), Germany). The system was fitted with a column, with dimensions of 30 m in length, 0.25 millimetres in internal diameter, and a film thickness of 0.25 micrometres. A sample of 1 microliter was introduced into the gas chromatography–mass spectrometry (GC–MS) system at a flow rate of 1 millilitre per minute, with helium serving as the carrier gas. The input temperature was set at 250 °C, while the transfer line temperature was set at 225 °C. The initial setting of the oven was set at 70 °C for a duration of 30 s. Subsequently, the temperature was decreased at a rate of 10 °C per minute until it reached 10 °C. It was then increased to 150 °C and maintained for a period of 2 min. Following this, the temperature was once again decreased at a rate of 10 °C per minute until it reached 10 °C. Finally, the temperature was raised to 330 °C for a duration of 180 s. The experimental settings suggested for the analysis of MS data included acquiring 13 spectra per second, a mass-to-charge ratio range of 30–1000 *m*/*z*, electron impact at an energy of 70 eV, and maintaining the ion source temperature at 250 °C and the extraction frequency at 1.25 Hz. Three sets of triplicate samples were subjected to three injections each, resulting in a total of nine injections per sample. The data were analysed with ChromaTOF^®^ program. All uses of active compounds and applications were gathered from NCBI PubChem database, which may be found at [37].

## 4. Conclusions

The findings of this study indicate that Aloe vera, a therapeutic plant, harbours a wide variety of bacteria such as *Enterobacter cloacae* I4 and *Pseudomonas fulva* I3–I5, which possess antimicrobial properties and significant biotechnological value. This study also found that endophytes create bioactive compounds which may be useful in sectors such as cosmetics, food, and beverages.

## Figures and Tables

**Figure 1 molecules-29-01297-f001:**
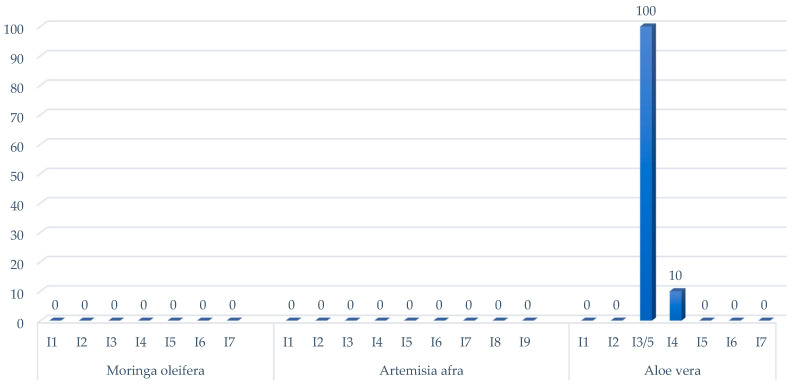
Bar graph showing the antimicrobial properties of the isolated bacteria endophytes from medicinal plants against *Mycobacterium bovis*. Twenty-one isolates showed no inhibition and two isolates showed inhibition, with isolate Aloe I3/5 exhibiting total inhibition. (0): no inhibition, (10): inhibition observed, (100): complete inhibition or no growth of *M. bovis* at all.

**Figure 2 molecules-29-01297-f002:**
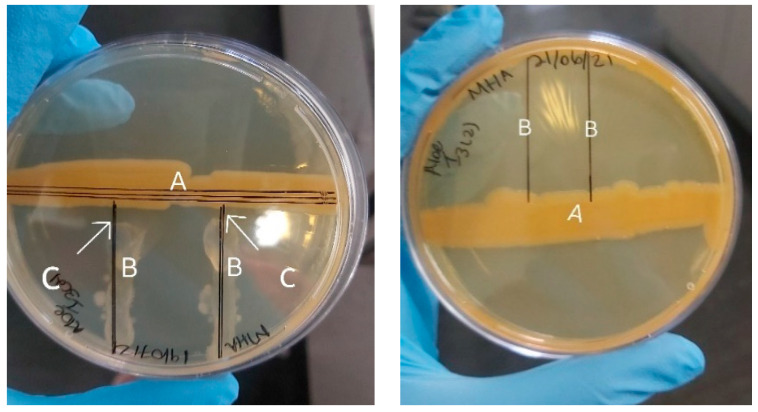
Photograph of the cross-streak method showing the antimicrobial activity of isolate Aloe I4 (**left**) with a 5 mm area of inhibition (A: Aloe I4, B: *M. bovis*; C: Zone of inhibition) and Aloe I3/5 (**right**) showing no growth of *M. bovis* after 5 days of incubation (A: Aloe I3–I5; B: *M. bovis*).

**Figure 3 molecules-29-01297-f003:**
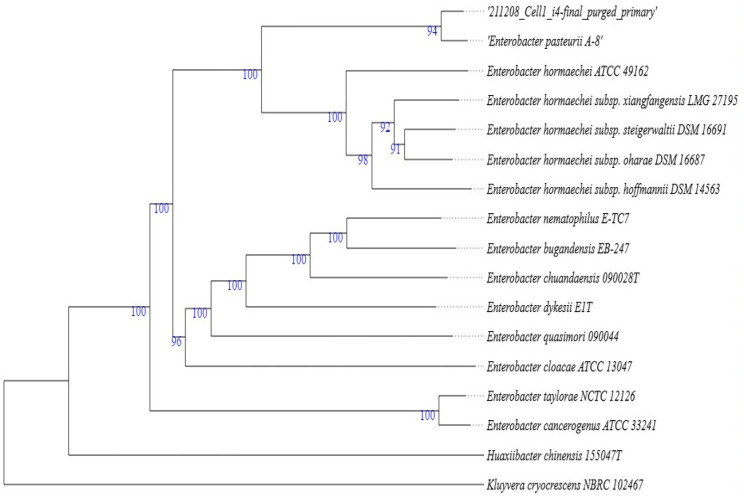
*Enterobacter cloacae* I4 phylogenetic tree based on de novo genome assembly analysis. The numbers beneath the branches represent bootstrap support levels based on 100 replications.

**Figure 4 molecules-29-01297-f004:**
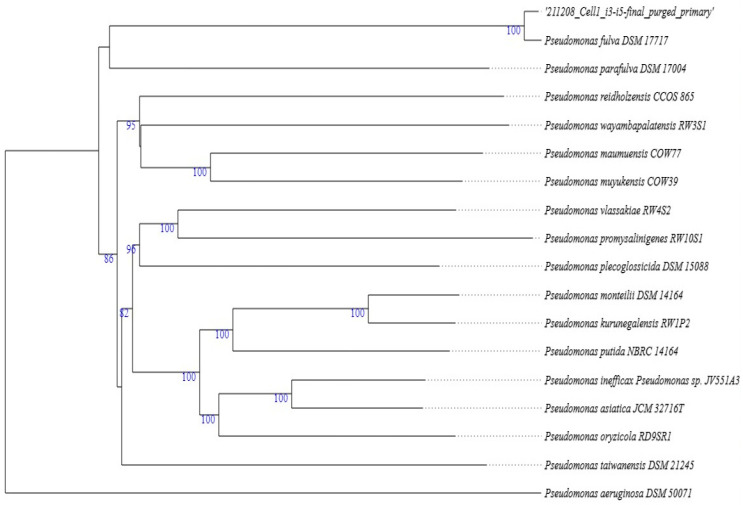
*Pseudomonas fulva* I3–I5 phylogenetic tree based on de novo genome assembly analysis. The numbers beneath the branches represent bootstrap support levels based on 100 replications.

**Table 1 molecules-29-01297-t001:** De novo genome assembly summary metrics for Aloe I4. Primary contigs represent pseudo haplotype assemblies, while haplotigs represent fully phased and assembled regions of the genome.

Contig Type	Polished Contigs	Maximum Contig Length	Mean Contig Length	Median Contig Length	N50 Contig Length	Sum of Contig Lengths	e-Size (Sum of Squares	Number of Circular Contigs
Primary contigs	3	2,598,250	1,572,706	1,434,460	2,598,250	4,718,120	1,966,539	0
Haplotigs	1	10.181	10.181	10.181	10.181	10.181	10.181	N/A

**Table 2 molecules-29-01297-t002:** De novo genome assembly summary metrics for Aloe I3–I5. Primary contigs represent pseudo haplotype assemblies, while haplotigs represent fully phased and assembled regions of the genome.

Contig Type	Polished Contigs	Maximum Contig Length	Mean Contig Length	Median Contig Length	N50 Contig Length	Sum of Contig Lengths	e-Size (Sum of Squares	Number of Circular Contigs
Primary contigs	1	4,840,834	4,840,834	4,840,834	4,840,834	4,840,834	4,840,834	1
Haplotigs	3	17.724	15.141	16.977	16.977	45.425	15.792	N/A

**Table 3 molecules-29-01297-t003:** Unique metabolites with varying abundances from *Pseudomonas fulva* I3–I5.

	R.T. (s)	Area%	Observed Ion *m*/*z*	Compound Name	Formula	Area	Similarity	Peak S/N
Alcohols								
1	738.27	0.035	206.166	2,4-Di-tert-butylphenol	C_14_H_22_O	117,980	888	650
2	371.83	0.96	122.07	Phenylethyl Alcohol	C_8_H_10_O	3,250,678	940	3197
3	763.75	0.14	125.06	3-Mercapto-3-methylbutanol	C_5_H_12_OS	479,614	785	813
4	957.02	0.07	161.08	Tryptophol	C_10_H_11_NO	224,164	752	328
Alkanes and Alkenes								
5	270.83	0.14	127.65	Tridecane, 4-methyl-	C14H30	454,419.5	905.5	488.0
6	789.12	1.54	196.04	Hexadecane	C_16_H_34_	5,258,816	940	3164
7	818.01	0.08	1671.612	2-Dodecanone	C_12_H_24_O	287,327	843	510
8	832.25	0.52	161.41	Pentadecane	C_15_H_32_	1,758,233	940	828
9	385.94	0.18	157.31	Tridecane	C_13_H_28_	603,935	922	821
10	1011.24	0.05	189.44	Octadecane, 4-methyl-	C_19_H_40_	176,886	927	174
11	1085.77	0.0475	130.6396	1-Iodo-2-methylundecane	C_12_H_25_I	156,952	934	136.5
12	1111.10	1.06	263.30	Eicosane	C_20_H_42_	3,600,474	965	1270
13	1328.84	0.14	196.89	Octacosane	C_28_H_58_	480,809	954	243
14	1225.79	0.62	239.27	Tetracosane	C_24_H_50_	2,113,643	958	801
15	1098.44	0.16	216.14	5-Eicosene, (*E*)-	C_20_H_40_	531,752	842.50	353.50
16	1107.145	0.13	182.14295	3-Eicosene, (*E*)-	C_20_H_40_	432,411	954.5	232.5
17	743.364	0.197	206.1307	7-Hexadecene, (*Z*)-	C_16_H3_2_	666,803	944	439
Esters								
18	421.96	0.03	150.07	Benzeneacetic acid, methyl ester	C_9_H_10_O_2_	97,981	908	204
19	1061.66	0.0275	186.561	1,2-Benzenedicarboxylic acid, butyl 2-ethylhexyl ester	C_20_H_30_O_4_	87,747.5	22,613.625	480
20	1067.12	0.03	135.07	Formic acid, 2-phenylethyl ester	C_9_H_10_O_2_	99,162	891	210
21	1080.37	0.02	292.20	Benzenepropanoic acid, 3,5-bis(1,1-dimethylethyl)-4-hydroxy-, methyl ester	C_18_H_28_O_3_	81,726	715	164
22	1061.69	0.03	187.06	Dibutyl phthalate	C_16_H_22_O_4_	98,061	916	275
23	1059.04	0.30	219.63	Ethyl 2-cyano-3-(4-methacryloyloxyphenyl)acrylate	C1_6_H_15_NO_4_	1,011,529	866	1293
Ketones								
24	916.74	0.04	169.09	3-Methyl-1,4-diazabicyclo[4.3.0]nonan-2,5-dione, N-acetyl-	C_10_H_14_N_2_O_3_	143,741	758	493
25	951.08	0.04	154.07	Pyrrolo[1,2-a]pyrazine-1,4-dione, hexahydro-	C_7_H_10_N_2_O_2_	145,827	848	459
26	1045.79	0.38	183.43	Pyrrolo[1,2-a]pyrazine-1,4-dione, hexahydro-3-(2-methylpropyl)-	C_11_H_18_N_2_O_2_	1,302,535	845	995
27	374.04	0.15	145.78	4-Piperidinone, 2,2,6,6-tetramethyl-	C_9_H_17_NO	514,662	758	1406
Other organic compounds								
28	167.99	8.01	56.54	Acetic acid, hydroxy-	C_2_H_4_O_3_	27,263,571	869	1676
29	273.27	0.01	121.09	Pyridine, 2,4,6-trimethyl-	C_8_H_11_N	6216	808	122
30	611.31	0.01	131.07	Indole, 3-methyl-	C_9_H_9_N	37,823	881	200
31	782.83	0.04	183.10	Dodecanoic acid	C_12_H_24_O_2_	111,574	868	186
32	1087.75	0.04	228.20	n-Hexadecanoic acid	C_16_H_32_O_2_	124,987	852	159
33	1215.02	0.10	211.11	l-Leucyl-d-leucine	C_12_H_24_N_2_O_3_	312,139	717	322
34	1320.19	0.10	244.62	Ergotaman-3′,6′,18-trione, 9,10-dihydro-12′-hydroxy-2′-methyl-5′-(phenylmethyl)-, (5′a,10a)-	C_33_H_37_N_5_O_5_	351,941	852	643

**Table 4 molecules-29-01297-t004:** Unique metabolites with varying abundances from *Enterobacter cloacae* I4.

No	R.T. (s)	Area%	Observed Ion *m*/*z*	Name	Formula	Area	Similarity	Peak S/N
Alcohols								
1	966.5	0.11	146.8	(3*S*,6*S*)-3-Butyl-6-methylpiperazine-2,5-dione	C_9_H_16_N_2_O_2_	913,414	723	559
2	697.2	0.03	155.1	1-Dodecanol	C_12_H_26_O	285,215	919	140
3	623.9	0.03	155.1	1-Undecanol	C_11_H_24_O	287,760	871	138
4	951.2	0.07	211.2	2,4-Di-tert-butylphenol	C_14_H_22_O	536,471	802	366
5	763.4	0.09	127.6	1-Octen-4-ol	C_8_H_16_O	270,020	790	401
6	520.0	0.05	128.0	2-Ethyl-1-hexanol	C_8_H_18_O	143,985	839	192
7	786.7	0.03	138.2	2-Undecen-4-ol	C_11_H_22_O	111,899	795	403
8	763.5	0.07	121.1	3-Mercapto-3-methylbutanol	C_5_H_12_OS	286,892	789	623
9	974.8	0.03	126	4-Mercaptophenol	C_6_H_6_OS	236,549	758	244
10	512.7	0.01	137.7	Ethanol, 1-methoxy-, benzoate	C_10_H_12_O_3_	126,066	778	138
11	606.2	0.79	166.1	Ethanol, 2-(2-butoxyethoxy)-	C_8_H_18_O_3_	6,801,240	886	17,274
12	646.6	0.26	141.4	Ethanol, 2-(2-butoxyethoxy)-, acetate	C_10_H_20_O_4_	2,226,963	918	1609
13	281.0	0.10	117.2	Ethanol, 2-(2-ethoxyethoxy)-	C_6_H_14_O_3_	650,282	833	365
14	942.1	0.04	133.6	Ethanol, 2-[2-(ethenyloxy)ethoxy]-	C_6_H_12_O_3_	306,041	787	227
15	164.1	0.31	48.0	Ethanol, 2,2-dichloro-	C_2_H_4_C_l2_O	967,422	947	142
16	269.7	8,42	32.0	Methyl Alcohol	CH_4_O	16,979,127	926	689
17	1161.2	0.04	139.8	n-Tetracosanol-1	C_24_H_50_O	166,939	936	108
18	1393.3	0.04	368.3	Phenol, 2,2′-methylenebis[6-(1,1-dimethylethyl)-4-ethyl-	C_25_H_36_O_2_	323,970	889	1769
19	738.2	0.02	206.2	Phenol, 2,5-bis(1,1-dimethylethyl)-	C_14_H_22_O	34,536	890	161
20	1246.6	0.17	228.1	Phenol′, 4,4′-(1-methylethylidene)bis-	C_15_H_16_O_2_	1,403,098	726	378
21	371.7	0.08	122.1	Phenylethyl Alcohol	C_8_H_10_O	455,009	899	653
Alkanes and Alkenes								
22	1258.1	0.01	259.1	1,3-Dioxolane-2-heptanenitrile, a-methyl-d-oxo-2-phenyl-	C_17_H_21_NO_3_	65,288	747	150
23	988.2	0.48	124.4	1,5-Heptadiene, 3,4-dimethyl-	C_9_H_16_	4,164,767	869	1555
24	1206.7	0.04	144.6	1,3-Cyclopentadiene, 5-(trans-2-ethyl-3-methylcyclopropylidene)-	C_11_H_14_	332,887	723	520
25	877.1	0.02	169.2	1-Docosene	C_22_H_44_	186,631	874	195
26	296.7	0.05	100.6	2-Decene, 5-methyl-, (*Z*)-	C_11_H_22_	55,377	848	128
27	825.5	0.03	172.6	3-Eicosene, (*E*)-	C_20_H_40_	51,075	759	166
28	708.5	0.06	182.1	3-Tetradecene, (*Z*)-	C_14_H_28_	178,811	932	242
29	964.4	0.12	158.1	4-Heptafluorobutyryloxyhexadecane	C_20_H_33_F_7_O_2_	346,559	941	255
30	673.6	0.07	143.3	5-Esene, (*E*)-	C_20_H_40_	269,736	806	316
31	812.1	0.18	150.4	7-Hexadecene, (*Z*)-	C_16_H_32_	487,592	940	378
32	974.1	0.09	175.2	9-Eicosene, (*E*)-	C_20_H_40_	766,434	909	245
33	1057.7	1.34	237.9	Eicosane	C_20_H_42_	2,954,435	959	1373
34	726.1	0.04	123.4	Heptadecane, 2,6,10,14-tetramethyl-	C_21_H4_4_	141,931	900	122
35	956.2	0.05	126.8	Heptadecane, 2-methyl-	C_18_H_38_	161,042	929	160
36	1625.0	0.08	211.4	Heptaethylene glycol	C_14_H_30_O_8_	209,475	930	166
37	1452.7	0.08	527.5	Heptasiloxane, hexadecamethyl-	C_16_H_48_O_6_Si_7_	690,174	758	417
38	794.7	0.84	171.9	Hexadecane	C_16_H_34_	1,956,218	943	1681
39	1486.3	0.07	159.4	Hexaethylene glycol	C_12_H_26_O_7_	196,923	898	173
40	1348.6	0.21	182.9	Octacosane	C_28_H_58_	509,942	953	264
41	931.3	0.06	156.5	Octadecane, 4-methyl-	C_19_H_40_	135,195	898	164
42	1237.8	0.76	247.1	Tetracosane	C_24_H_50_	2,390,307	955	855
43	548.8	0.28	153.0	Tridecane	C_13_H_28_	712,424	918	798
44	868.7	0.04	127.1	Tridecane, 4-methyl-	C_14_H_30_	116,341	907	130
Esters								
45	1005.1	0.06	143.4	1,2-Benzenedicarboxylic acid, butyl 2-ethylhexyl ester	C_20_H_30_O_4_	523,317	727	365
46	1092.4	0.03	150.0	1,2-Benzenedicarboxylic acid, dipropyl ester	C_14_H_18_O_4_	97,603	886	539
47	1546.4	0.97	294.5	1,2-Benzenedicarboxylic acid, decyl octyl ester	C_26_H_42_O_4_	715,862	893	152
48	1056.9	0.01	183.4	1H-Indole-3-ethanol, acetate (ester)	C_12_H_13_NO_2_	88,047	829	250
49	1377.6	0.01	195.0	2-Fluoro-3-trifluoromethylbenzoic acid, 3-methylbutyl-2 ester	C_13_H_14_F_4_O_2_	92,029	781	513
50	904.9	0.03	149.5	2-Propenoic acid, 2-methyl-, oxiranylmethyl ester	C_7_H_10_O_3_	107,925	792	425
51	668.6	0.04	126.8	2-Propenoic acid, tridecyl ester	C_10_H_18_N_2_O_2_	115,873	798	381
52	1308.8	0.01	273.5	2,2-diphenylpropionic acid, 2,2,2-trifluoroethyl ester	C_17_H_15_F_3_O_2_	101,730	743	173
53	266.8	0.18	86.	Acetic acid ethenyl ester	C_4_H_6_O_2_	437,551	975	312
54	754.1	0.01	194.1	Benzoic acid, 4-ethoxy-, ethyl ester	C_11_H_14_O_3_	82,696	868	365
55	1563.6	0.04	180.0	Carbonic acid, nonyl vinyl ester	C_12_H_22_O_3_	339,969	876	107
56	765.8	0.14	202.2	Cyclopropanecarboxylic acid, 2-ethylhexyl ester	C_12_H_22_O_2_	1,233,173	811	1009
57	1132.1	0.05	154.1	Di(1-methylcyclobutyl) ether	C_10_H_18_O	370,096	814	213
58	819.8	0.01	177.7	Diethyl Phthalate	C_12_H_14_O_4_	112,334	938	539
59	1404.6	0.06	204.6	Diisooctyl phthalate	C_24_H_38_O_4_	67,422	860	340
60	1088.3	1.33	215.4	DL-Alanine, *N*-methyl-*N*-(byt-3-yn-1-yloxycarbonyl)-, tetradecyl ester	C_23_H_41_NO_4_	11,535,230	890	817
61	892.0	0.08	186.5	Hexanedioic acid, bis(2-methylpropyl) ester	C_14_H_26_O_4_	649,583	900	3117
62	1398.2	0.39	282.6	l-Norvaline, n-propargyloxycarbonyl-, nonyl ester	C_18_H_31_NO_4_	1,742,412	909	232
63	1039.5	0.64	192.1	*L*-Proline, *N*-valeryl-, decyl ester	C_20_H_37_NO_3_	5,550,110	788	850
64	1568.0	0.09	237.7	Oxalic acid, allyl decyl ester	C_15_H_26_O_4_	809,220	907	186
65	1356.1	0.53	363.7	Phosphoric acid, isodecyl diphenyl ester	C_22_H_31_O_4_P	4,518,474	881	5661
66	1367.5	0.03	184.1	Phosphoric acid, tris(2-ethylhexyl) ester	C_24_H_51_O_4_P	211,680	804	951
67	1559.3	0.35	293.5	Phthalic acid, 4-chloro-2-methylphenyl tetradecyl ester	C_29_H_39_C_l_O_4_	381,354	863	111
68	1544.1	0.86	294.2	Phthalic acid, 7-methyloct-3-yn-5-yl undecyl ester	C_28_H_42_O_4_	771,078	861	152
69	1472.5	0.66	321.1	Phthalic acid, 8-chlorooctyl decyl ester	C_26_H_41_ClO_4_	1,124,586	901	2078
70	841.5	0.07	192.1	Propanoic acid, 2-methyl-, 2-phenylethyl ester	C_12_H_16_O_2_	564,174	734	532
71	0.00	0.001	515.4	Propanoic acid, 3,3′-thiobis-, didodecyl ester	C_30_H_58_O_4_S	456,850	748	487
72	1064.7	0.01	227.2	Tridecanoic acid, methyl ester	C_14_H_28_O_2_	61,779	814	144
73	1065.2	0.01	206.6	Undecanoic acid, methyl ester	C_12_H_24_O_2_	98,910	813	135
Ketones								
74	1166.8	0.04	225.1	1,3-Propanedione, 2-bromo-1,3-diphenyl-	C_15_H_11_BrO_2_	320,260	858	575
75	1043.6	0.01	140.1	1-(2-Thienyl)-1-propanone	C_7_H_8_OS	133,221	731	172
76	921.0	0.16	205.1	2,2-Dimethyl-*N*-phenethylpropionamide	C_13_H_19_NO	1,219,924.7	792.4	786.5
77	1233.1	0.0	204.1	2,5-Piperazinedione, 3-(phenylmethyl)-	C_11_H_12_N_2_O_2_	146,740	839	110
78	472.4	0.05	134.0	2-Coumaranone	C_8_H_6_O_2_	154,354	877	274
79	743.4	0.06	155.5	2-Dodecanone	C_12_H_24_O	232,592	831	256
80	432.2	0.05	104.0	2-Pentanone, 4-hydroxy-4-methyl-	C_6_H_12_O_2_	123,821	804	337
81	381.7	0.03	143.1	4-Butoxy-2-butanone	C_8_H_16_O_2_	76,347	752	153
82	373.5	0.10	145.8	4-Piperidinone, 2,2,6,6-tetramethyl-	C_9_H_17_NO	419,489	775	613
83	1018.1	0.31	154.1	5-Pyrrolidino-2-pyrrolidone	C_8_H_14_N_2_O	1,355,861	709	477
84	1501.5	0.03	116.9	Acetone, 1-[4-(dimethylaminoethoxy)phenyl]-	C_13_H_19_NO_2_	52,377	925	258
85	218.6	0.02	94.0	Dimethyl sulfone	C_2_H_6_O_2_S	63,528	889	472
86	188.0	0.02	78.0	Dimethyl Sulfoxide	C_2_H_6_OS	54,269	781	147
87	910.8	0.07	185.1	Cyclopenta[c]quinolin-4-one, 1,2,3,5-tetrahydro-	C_12_H_11_NO	110,533	807	574
88	1283.8	0.01	260.1	Cyclopentanone, 2,5-bis(phenylmethylene)-	C_19_H_16_O	33,095	808	182
89	1076.1	0.11	218.0	Diphenyl sulfone	C_12_H_10_O_2_S	285,983	818	1552
90	956.1	0.05	154.1	Pyrrolo[1,2-a]pyrazine-1,4-dione, hexahydro-	C_7_H_10_N_2_O_2_	135,989	880	114
91	1048.4	2.08	203.6	Pyrrolo[1,2-a]pyrazine-1,4-dione, hexahydro-3-(2-methylpropyl)-	C_11_H_18_N_2_O_2_	5,818,832	819	2202
Other organic compounds								
92	857.6	0.01	161.1	3-Benzyl-5-chloro-1,2,3-triazole 1-oxide	C_9_H_8_C_l_N_3_O	63,591	802	121
93	1112.1	0.02	241.1	Metolachlor	C_15_H_22_C_l_NO_2_	199,899	775	1110
94	793.3	0.05	184.1	3-(2-Phenylethyl)pyridazine	C_12_H_12_N_2_	38,560	716	211
95	1121.6	0.10	475.0	3-Isopropoxy-1,1,1,7,7,7-hexamethyl-3,5,5-tris(trimethylsiloxy)tetrasiloxane	C_18_H_52_O_7_Si_7_	835,937	718	824
96	917.8	0.09	168.7	3-Methyl-1,4-diazabicyclo[4.3.0]nonan-2,5-dione, N-acetyl-	C_10_H_14_N_2_O_3_	208,480	790	309
97	224.7	0.37	190.5	4-(2-Acetoxyphenyl)-1-ethyl-3-methyl-5-(4-nitrophenyl)pyrazole	C_20_H_19_N_3_O_4_	3,198,062	999	332
98	1105.6	0.03	182.1	9H-Pyrido[3,4-b]indole, 1-methyl-	C_12_H_10_N_2_	60,640	878	306
99	1051.0	0.05	235.2	Acetamide, 2-chloro-N-(ethoxymethyl)-N-(2-ethyl-6-methylphenyl)-	C_14_H_20_C_l_NO_2_	455,685	760	507
100	177.0	8.87	66.5	Acetic acid, hydroxy-	C_2_H_4_O_3_	39,385,471	788	2051
101	267.8	0.11	109.5	Acetic anhydride	C_4_H_6_O_3_	322,969	995	273
102	1215.3	0.08	222.1	Benzene, (1,2-dicyclopropyl-2-phenylethyl)-	C_20_H_22_	696,698	732	371
103	865.4	0.01	132.1	Benzene, (1-azido-1-methylethyl)-	C_9_H_11_N_3_	83,422	761	165
104	621.3	0.01	135.1	Benzeneacetamide	C_8_H_9_NO	100,313	846	217
105	519.7	0.04	146.1	Benzeneethanamine, N-(1-methylethylidene)-	C_11_H_15_N	125,968	879	135
106	651.1	0.09	165.5	Benzeneethanamine, N-(3-methylbutylidene)-	C_13_H_19_N	254,728	847	133
107	652.6	0.04	138.1	Benzeneethanol, 4-hydroxy-	C_8_H_10_O_2_	309,979	876	356
108	962.9	0.04	212.1	Benzyl Benzoate	C_14_H_12_O_2_	361,122	873	444
109	603.3	0.06	134.1	Bicyclo[3.1.0]hex-2-ene, 4-methylene-1-(1-methylethyl)-	C_10_H_14_	111,702	756	259
110	1309.0	0.02	273.6	Bifenthrin	C_23_H_22_C_l_F_3_O_2_	135,178	725	284
111	1406.2	0.76	390.6	Bis(2-ethylhexyl) phthalate	C_24_H_38_O_4_	6,611,090	900	5447
112	1416.0	0.03	315.1	Bumetrizole	C_17_H_18_C_l_N_3_O	302,918	943	1675
113	518.3	0.05	156.1	Butanamide, N-hexyl-	C_10_H_21_NO	465,622	724	421
114	181.5	2.30	97.0	Butanoic acid, 3-methyl-	C_5_H_10_O_2_	1,955,281	856	632
115	743.7	0.09	220.2	Butylated Hydroxytoluene	C_15_H_24_O	802,322	931	3077
116	964.6	0.10	137.4	Cyclo-(glycyl-l-leucyl)	C_8_H_14_N_2_O_2_	198,566	760	171
117	1536.4	0.04	86.1	Cyclobutane, methoxy-	C_5_H_10_O	164,364	891	431
118	877.9	0.09	437.0	Cyclooctasiloxane, hexadecamethyl-	C_16_H_48_O_8_Si_8_	791,787	719	1152
119	1361.3	0.17	287.5	Di-n-octyl phenyl phosphate	C_22_H_39_O_4_P	1,439,556	733	5036
120	961.9	0.27	168.8	dl-Alanyl-l-leucine	C_9_H_18_N_2_O_3_	2,322,399	853	247
121	1000.9	0.02	201.7	Dodecanoic acid, 2-methyl-	C_13_H_26_O_2_	210,456	857	317
122	860.5	0.03	172.6	Dodecanoic acid	C_12_H_24_O_2_	113,150	832	158
123	898.2	0.04	253.3	Dodecyl acrylate	C_15_H_28_O_2_	347,139	939	273
124	1330.3	2.56	291.3	Ergotaman-3′,6′,18-trione, 9,10-dihydro-12′-hydroxy-2′-methyl-5′-(phenylmethyl)-, (5′α,10a)-	C_33_H_37_N_5_O_5_	7,364,698	870	2348
125	340.0	0.04	103.6	Glycyl-dl-norvaline	C_7_H_14_N_2_O_3_	362,257	773	431
126	1665.0	0.02	432.1	Hexasiloxane, tetradecamethyl-	C_14_H_42_O_5_Si_6_	170,683	750	131
127	1648.3	0.03	342.3	Indeno[1,2-b]pyridine, 7-methyl-5-(2,2,6,6-tetramethylpiperid-4-ylimino)-	C_22_H_27_N_3_	270,671	866	1478
128	522.0	0.01	117.1	Indole	C_8_H_7_N	63,439	854	329
129	484.7	0.25	146.1	Isobutyramide, N-(3-methylbutyl)-	C_9_H_19_NO	2,176,992	765	1702
130	708.8	0.00	146.1	c, 1,2,3,4-tetrahydro-1,8-dimethyl-	C1_2_H_16_	14,327	729	124
131	1148.5	0.05	227.5	n-Hexadecanoic acid	C_16_H_32_O_2_	258,286	825	177
132	358.7	0.03	106.1	Nonanal	C_9_H_18_O	110,278	815	162
133	854.5	0.02	167.1	Octanamide, N,N-dimethyl-	C_10_H_21_NO	162,285	796	836
134	1352.4	0.06	252.0	Octicizer	C_20_H_27_O_4_P	46,610	810	122
135	1332.4	0.37	247.6	Oxamide, N-(3-m′thoxypropyl)-N′-cycloheptylidenamino-	C_13_H_23_N_3_O_3_	3,237,962	953	828
136	163.1	1.67	32.0	Oxygen	O_2_	5,306,645	957	488
137	339.0	0.05	103.6	Pentanoic acid	C_5_H_10_O_2_	424,336	741	250
138	832.1	0.05	191.1	Phenylacetamide, N-isobutyl-	C_12_H_17_NO	387,577	819	640
139	922.9	0.10	209.1	Phenylacetamide, N-pentyl-	C_13_H_19_NO	788,730	836	282
140	263.5	0.09	182.3	Phosphonic acid, (p-hydroxyphenyl)-	C_6_H_7_O_4_P	817,511	814	3495
141	1347.1	0.03	260.1	terphenyl, 4,4″-diamine	C_18_H_16_N_2_	77,775	817	414
142	793.4	0.03	184.1	Pyridine, 3-(4-tolylamino)-	C_12_H_12_N_2_	101,812	729	552
143	1144.4	0.02	184.8	Quinoline, 2-(2-methylpropyl)-	C_13_H_15_N	180,409	750	238
144	890.7	0.02	191.5	ß-Phenylethyl butyrate	C_12_H_16_O_2_	225,091	759	162
145	1344.8	0.12	326.1	Triphenyl phosphate	C_18_H_15_O_4_P	1,059,663	917	648
146	938.5	0.06	1878	Undecanoic acid	C_11_H_22_O_2_	351,308	852	235
147	537.7	0.34	16.8	Valeramide, N-hexyl-	C_11_H_23_NO	2,316,095	730	3538

## Data Availability

Data are contained within the article.

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
