# Peer review of "The Indiscriminate Chemical Makeup of Secondary Metabolites Derived from Endophytes Harvested from Aloe barbadensis Miller in South Africa’s Limpopo Region"

_molecules, 2024, doi:10.3390/molecules29061297_

Round 1
Reviewer 1 Report
Comments and Suggestions for Authors
Some comments for improvement
1. Title should be revised since in the contents showed the results of isolation endophytes from not only Aloe vera
2. Table 2 must be reformulated to give concise information
3. There's no Figure 1
4. Figure 2 & 3 should be revised, it can't be seen clearly
Author Response
The authors went through the reviewer's comments and addressed them point by point and the report can be seen in the attached report

Reviewer 2 Report
Comments and Suggestions for Authors
1. The English of the manuscript is very poor, the author should polish the English.
2. The author should rewrite the Abstract, because it is difficult to understand.
3. The 23 bacterial should all identified the genus.
4. “Two endophytes, Aloe I4 and Aloe I3-I5 from Aloe vera exhibited greater 17 potency, with inhibitory zones of 15-, and 20 mm diameter, respectively.” I can not find this content in the other parts of manuscript except abstract. Besides, the author should add the antibacterial activity all 23 bacterial. The author should also add the concentration and positive control.
5. Figures 2 and 3 is not clearly, the author should change the pictures. And I am not find the Figure 1.
6. The format of the References should revised, such as 3, 12, 16, 21.
Comments on the Quality of English LanguageThe English of the manuscript is very poor, the author should polish the English.
Author Response
Dear reviewer, the authors went through the comments and addressed them point by point and the report is attached below.

Round 2
Reviewer 2 Report
Comments and Suggestions for Authors
The paper can be accepted.